



# Gradient-based Wind Farm Layout Optimization With Inclusion And Exclusion Zones

Javier Criado Risco[1], Rafael Valotta Rodrigues[1], Mikkel Friis-Møller[1], Julian Quick[1], Mads Mølgaard Pedersen[1], and Pierre-Elouan Réthoré[1]

[1]Technical University Of Denmark, Frederiksborgvej 399, 4000 Roskilde, Denmark

**Correspondence:** Javier Criado Risco (jcrri@dtu.dk)

**Abstract.** Wind farm layout optimization is usually subjected to boundary constraints of irregular shapes. The analytical expressions of these shapes are rarely available, and consequently, it can be challenging to include them in the mathematical formulation of the problem. This paper presents a new methodology to integrate multiple disconnected and irregular domain boundaries in wind farm layout optimization problems. The method relies on the analytical gradients of the distances between wind turbine locations and boundaries, which are represented by polygons. This parameterized representation of boundary locations allows for a continuous optimization formulation. A limitation of the method, if combined with gradient-based solvers, is that wind turbines are placed within the nearest polygons when the optimization is started in order to satisfy the boundary constraints, thus the allocation of wind turbines per polygon is highly dependent on the initial guess. To overcome this and improve the quality of the solutions, two independent strategies are proposed. A study case is presented to demonstrate the applicability of the method and the proposed strategies. In this study, a wind farm layout is optimized in order to maximize the AEP in a non-uniform wind resource site. The problem is constrained by the minimum distance between wind turbines and five irregular polygon boundaries, defined as inclusion zones. Initial guesses are used to instantiate the optimization problem, which is solved following three independent approaches: (1) a baseline approach that uses a gradient-based solver, (2) approach 1 combined with the relaxation of the boundaries, which allows for a better design space exploration, and (3) the application of a heuristic algorithm, *smart-start*, prior to the gradient-based optimization, improving the allocation of wind turbines within the inclusion polygons based on the potential wind resource and the available area. The results show that the relaxation of boundaries combined with a gradient-based solver achieves on average +10.2% of AEP over the baseline, whilst the *smart-start* algorithm, combined with a gradient-based solver, finds on average +20.5% of AEP with respect to the baseline and +9.4% of AEP with respect to the relaxation strategy.

## 1 Introduction

Wind farm layout design is usually subjected to geometric constraints, which can be dictated by the seabed conditions, water depth or local maritime routes in offshore projects, or by land ownership, presence of other infrastructure or existence of humid areas and waterways in onshore projects (Dalla Longa et al., 2018). An ideal configuration would consist of a single regular





and convex polygon within which all the wind turbines are placed. However, developers usually have to deal with multiple
complex and non-connected shapes that complicate the farm design phase.

When irregular, disconnected and non-convex shaped polygons are involved, the wind farm layout optimization framework
becomes challenging, as it is not straightforward to include analytical expressions of these areas in the problem formulation.
Despite a lot of research work having been done in the field of wind farm layout optimization, less attention has been given to
the implementation of complex shaped boundaries. Mittal and Mitra (2019) pointed out the "gap towards the development of
methods that lead to high quality solutions while considering constraints such as forbidden areas". Reddy (2021) discusses the
lack of "a robust method for modeling irregular, non-convex and disconnected domains".

Much of the prior work in optimizing wind farms with irregular boundaries has focused on discrete parametrization of the
domain and polygon representation to handle the constraints. Perez-Moreno et al. (2018) dealt with the preliminary design of
the turbine layout, electrical collection system, and support structures following first a sequential and then a multidisciplinary
approaches. For these purposes, they divided the design space in squares to enable the optimization with a Particle Swarm Op-
timization (PSO) algorithm. González et al. (2017) compared two meta-heuristic optimization algorithms (Genetic Algorithm
and Particle Swarm Optimization) to solve the layout optimization problem considering realistic constraints for the concession
zone and the maximum area. For their work, they discretized their domain as an analytical expression for these constraints was
not available. Chen and MacDonald (2014) involved land ownership participation rates in the wind farm layout optimization
problem. They split their domain in cells in order to account for the different land owners. Tao et al. (2021) shows an optimiza-
tion example with two restriction zones (an ellipse and a regular convex polygon), with straight-forward analytical expressions
due to the simple shapes. The whole domain is later discretized before the optimization is run, excluding the grid points that
belong to the forbidden zones. Mittal and Mitra (2019) solved a multiobjective wind farm optimization problem with restricted
areas within their domain by discretizing the area into a uniform set of grids. Shakoor et al. (2016) approached the problem
using definite point selection, considering an irregular polygon exclusion zone within a discretized domain. Sorkhabi et al.
(2018) present a continuous genetic algorithm optimization approach, only discretizing the domain when handling exclusion
zone constraints.

A disadvantage of discretizing the domain is that there is a trade-off between the quality of the solution and the resolution
of the grid. A thinner grid will lead a higher quality solution, whilst it will also increase significantly the computational effort
(Mittal and Mitra, 2019; Masoudi and Baneshi, 2022). Other strategies to integrate different boundaries include dividing the
domain into polygons and/or applying penalty functions. Sorkhabi et al. (2016) considered restricted areas defined by means of
convex polygon representation. The polygon vertices are joined with the wind turbine positions, forming triangles. If the area
of the triangles is equal to the area of the polygon, the turbine is located inside the polygon. Then the constraints are integrated
within the optimization framework by means of penalty functions. Feng et al. (2018) solved the AEP maximization problem in
complex terrain with convex shaped polygon boundary constraints. Their method, however, cannot handle irregular non-convex
boundaries. Hou et al. (2016) developed a layout optimization using Particle Swarm Optimization (PSO) considering two
restriction zones in a construction sea area, corresponding to an oil well and a gas pipe/Marine traffic lane. The implementation
associated with the restriction areas used a penalty function, which required a penalty factor value assumption. Optimized



solutions will, though, be sensitive to the choice for the penalty factor parameter. Similarly, the work done by Afanasyeva
et al. (2018) considered exclusion zones through the use of a penalty function. Wang et al. (2015) presents a genetic algorithm
optimization approach for wind farm layouts where exclusion zones are filtered as a pre-processing step. Another approach
was described in Reddy (2021), where a solution for modeling irregular, disconnected, complex and non-convex polygons
is proposed. This methodology aims at simplifying complex domain boundaries and land constraints with the vector domain
description (SVDD) technique. The SVDD is used to convert the regions into a space where the complex domain boundaries
can be represented as a spherical boundary, without compromising the accuracy of the optimization. This solution is more
advanced but also requires training the model with a sufficient amount of samples.

In this article, we propose a new methodology to integrate multiple irregular, non-convex and disconnected boundary con-
straints into the wind farm layout optimization problem. The method relies on polygon representation, given by their vertices.
The distance from every wind turbine to the polygons can be efficiently calculated by a set of geometric formulas that deter-
mine the nearest boundary and the sign of the distance towards it. Based on the sign of this distance, it is always possible to
identify if the wind turbine is inside or outside the considered polygon.

When this framework is used with gradient-based optimization, the wind turbines are placed within one of the inclusion
zones around them within the first iterations, since the solver will try to satisfy all the constraints when the optimization is
started. This means that the solution will be highly dependent on the initial positions. Additionally, if our inclusion zones
consist of many polygons spread across the design space, conventional gradient-based optimization using multiple random
starts may not be an effective approach to design wind farm layouts that fully utilize the available wind resource.

We have contemplated two possible solutions to overcome this challenge. The first solution is to introduce a relaxation term
in the boundary constraint formulation that diminishes during the optimization. This is achieved by adding an offset to the
distance determined by the method. In other words, the inclusion zones are expanded such that more of the domain can be
explored. The relaxation linearly decays during a number of optimization iterations, while the boundaries gradually return
to the true geometry. The second solution is the application of a heuristic algorithm, the *smart-start* (Rodrigues et al., 2023
forthcoming), which takes a discretized grid covering the domain as input, removes all points outside the inclusion zones and
then iteratively adds turbines one by one. In each iteration the wake deficit from already added turbines is calculated and the
next turbine is placed at the position with highest power potential.

The presented framework has been implemented in TOPFARM, the Technical University of Denmark (DTU) open source
software for wind farm optimization (Réthoré et al., 2014; DTU Wind Energy Systems, 2022b). A case study is presented,
where the three introduced approaches are followed to maximize the Annual Energy Production (AEP) of a wind farm in
complex terrain with several irregularly shaped and disconnected inclusion zones. In this study, a gradient-based driver is
combined with the relaxation of boundaries and with the *smart-start* algorithm to demonstrate the applicability of the method
and how the aforementioned challenge can be overcome. For the AEP calculation and the wake modeling involved in the
simulations, TOPFARM relies in PyWake (Pedersen et al., 2019), another DTU open source Python library that offers fast
AEP evaluation from a range of engineering wake models. Recent works using TOPFARM (Ciavarra et al., 2022; Riva et al.,





2020), and PyWake (Rodrigues et al., 2022; Fischereit et al., 2021; Forsting et al., 2021; Pedersen et al., 2021; Quick et al., 2022) can be found in the literature.

The article is structured as follows: Section 2 describes the mathematical principles and formulation of the method, including the boundary relaxation, the idea behind the *smart-start*, and how the flow is modeled along this work. Section 3 introduces the study case, describing in detail each of the approaches used, and providing a relaxation study which was used to decide the suitable parameters for the second approach. Section 4 presents the results and discussion. Eventually, Section 5 summarizes the conclusions and points at future work.

## 2   Methods

Given a set of wind turbines, $I$, we wish to maximize the Annual Energy Production (AEP) by finding the optimal layout of the farm. Our problem is constrained by a minimum distance between each pair of wind turbines, and several boundary constraints given as a set of disconnected polygons that are defined as inclusion or exclusion zones (i.e., the areas where the wind turbines are allowed or not allowed to be placed respectively). Each polygon is formed by a number of boundary edges.

This optimization problem is mathematically formulated as:

$$\max_{\boldsymbol{x},\boldsymbol{y}} \quad AEP(\boldsymbol{x},\boldsymbol{y})$$
$$\text{s.t.} \sqrt{\left(x_i - x_j\right)^2 + \left(y_i - y_j\right)^2} \geq S_{\min}, \quad \forall i,j \in I : i \neq j$$
$$C_i \geq 0, \qquad\qquad\qquad \forall i \in I \tag{1}$$
$$x_{\min} < x_i < x_{\max}, \qquad\qquad \forall i \in I$$
$$y_{\min} < y_i < y_{\max}, \qquad\qquad \forall i \in I$$

where $\boldsymbol{x}$ and $\boldsymbol{y}$ are the wind turbine coordinate vectors, $x_{\min}, x_{\max}, y_{\min}$ and $y_{\max}$ are the lower and upper limits for the design variables respectively, $S_{\min}$ is the minimum distance between turbines, and the term $C_i$ represents the signed distance from a wind turbine $i$ towards the nearest boundary edge from the polygon set. In this context, signed distance means that if $C_i$ is

positive, the wind turbine is inside an inclusion zone or outside an exclusion zone, whereas if it is negative, the wind turbine is inside an exclusion zone or outside an inclusion zone.

### 2.1   Distance to nearest polygon

In the aforementioned optimization problem (Eq. 1), a set of wind turbines and a set of polygons were given. The wind turbine locations are defined by their coordinates, and the polygons are defined by the coordinates of their vertices. The polygons can

be split into boundary edges. Hence, any pair of adjacent vertices belonging to the same polygon form a boundary edge. For example, two adjacent vertices of a polygon, with coordinates $(x'_k, y'_k)$ and $(x'_{k+1}, y'_{k+1})$ respectively, will form a boundary edge $k$, represented by a vector $\boldsymbol{e}_k$, whose components are defined as $(x'_{k+1} - x'_k, y'_{k+1} - y'_k)$. The polygons follow a hierarchy



that allows for the imposition of an exclusion zone on top of an inclusion zone and vice-versa. If more than one polygon of the same type overlap, these are merged into a single one, where the interceptions between boundary edges form the new vertices.

The main idea behind the method is to identify the nearest boundary edge from the wind turbine locations, calculate the distance to the nearest point on the respective boundary edge, and then compute the gradients of those distances with respect to the turbine locations to indicate the path that will lead them to the permitted areas. For this, a sequential approach is followed:

1. For each wind turbine, identify the nearest point on all boundary edges.

2. Compute the signed distances between each wind turbine and the identified nearest points, where positive distances mean
inside an inclusion zone or outside an exclusion zone.

3. Identify the nearest edge (and polygon) by finding the minimum of the previously calculated signed distances, in absolute value.

4. Calculate gradients of the signed distance with respect to the wind turbine positions.

Besides, for each boundary edge, we define a normal unitary vector that points inside the inclusion zone polygons and
outside the exclusion zone polygons, as illustrated in Figure 1. The purpose of the normal vectors is to indicate the correct side of the edge (where to place the turbines), and they are used to calculate the sign of the distances.

In order to mathematically formulate the method, we will demonstrate how to calculate the distance from a wind turbine $i$ to an inclusion zone polygon of $N$ vertices. The wind turbine location is defined by its coordinates $(x_i, y_i)$ and the polygon is defined by its vertex coordinates $(x'_1, y'_1), (x'_2, y'_2), \ldots (x'_{N-1}, y'_{N-1}), (x'_N, y'_N)$. The polygon can be split in $N$ boundary edges,
$\boldsymbol{e}_1, \boldsymbol{e}_2, ..., \boldsymbol{e}_N$. The last vector, $\boldsymbol{e}_N$, is defined from the last vertex, $(x'_N, y'_N)$, to the first one, $(x'_1, y'_1)$, closing the polygon. In addition, for every boundary edge, we define the unitary normal vectors, $\boldsymbol{n}_1, \boldsymbol{n}_2, ... \boldsymbol{n}_N$, that point to the interior of the polygon, as it represents an inclusion zone (see Figure 2).

The first step is to identify the nearest point from the wind turbine on all boundary edges. For this purpose, we define $\boldsymbol{a}_{ik}$ as the vector from the first vertex of the boundary edge $\boldsymbol{e}_k$ to the wind turbine location, and $\boldsymbol{b}_{ik}$ as the vector from the second
vertex of the boundary edge $\boldsymbol{e}_k$ to the wind turbine location (notice that $\boldsymbol{b}_{ik}$ is equivalent to $\boldsymbol{a}_{i,k+1}$). Now we can calculate $\tilde{a}_{ik}$, which is the projection of $\boldsymbol{a}_{ik}$ into $\boldsymbol{e}_k$ (Eq. 2), and $\hat{a}_{ik}$, which is the projection of $\boldsymbol{a}_{ik}$ into $\boldsymbol{n}_k$ (4):

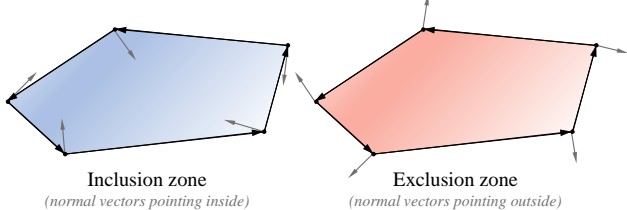

Inclusion zone
(normal vectors pointing inside)       Exclusion zone
(normal vectors pointing outside)

**Figure 1.** Definition of the polygon depending on the inclusion or exclusion attribute. **Left**: normal unitary vectors pointing inside the polygon, representing an inclusion zone; **Right**: normal unitary vectors pointing outside the polygon, representing an exclusion zone.





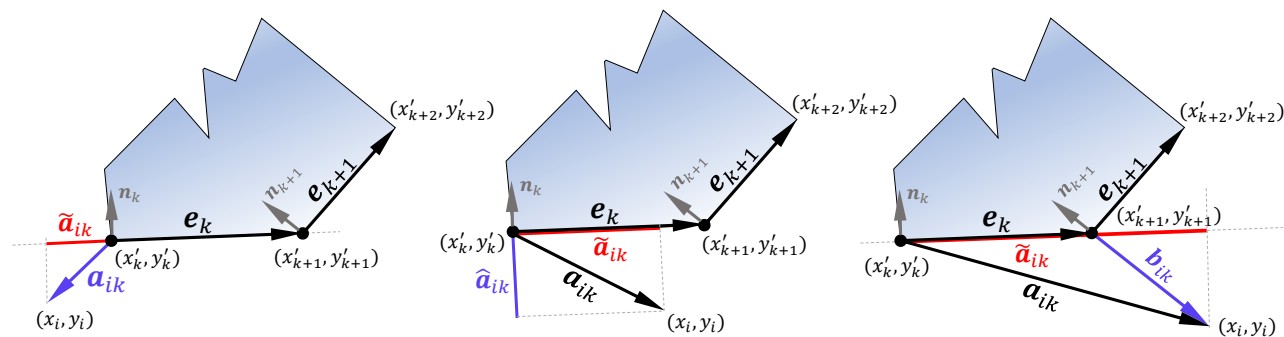

**Figure 2. Left**: wind turbine $i$ is closer to the first vertex of the boundary edge $\boldsymbol{e}_k$ (1); **Middle**: wind turbine $i$ is closer to an intermediate point of the boundary edge $\boldsymbol{e}_k$ (2); **Right**: wind turbine $i$ is closer to the second vertex of the boundary edge $\boldsymbol{e}_k$ (3). The red lines, $\tilde{a}_{ik}$, correspond to the projection of $\boldsymbol{a}_{ik}$ on boundary $\boldsymbol{e}_k$, while the blue lines/vectors correspond to the shortest distance between turbine $i$ and the boundary edge $\boldsymbol{e}_k$.

$$\tilde{a}_{ik} = \frac{\boldsymbol{a}_{ik} \cdot \boldsymbol{e}_k}{|\boldsymbol{e}_k|} \tag{2}$$

$$\hat{a}_{ik} = \boldsymbol{a}_{ik} \cdot \boldsymbol{n}_k \tag{3}$$

Depending on the relative position of the wind turbine with respect to the boundary edge $\boldsymbol{e}_k$, we can distinguish three

possibilities based on $\tilde{a}_{ik}$: (1) if $\tilde{a}_{ik}$ is negative, the wind turbine is closer to the first vertex of the edge $\boldsymbol{e}_k$; (2) if $\tilde{a}_{ik}$ is positive and less than or equal to the length of the boundary edge, the wind turbine is closer to an intermediate point of the edge $\boldsymbol{e}_k$; and (3) if $\tilde{a}_{ik}$ is positive and larger than the length of the boundary edge, the wind turbine is closer to the second vertex of the edge $\boldsymbol{e}_k$. Figure 2 illustrates the different scenarios.

Once the nearest point on each boundary edge has been identified, the second step consists of computing the shortest signed

distance between the wind turbine and the boundary edge. For case (2), this corresponds with the perpendicular distance, $\hat{a}_{ik}$. For cases (1) and (3), we need an additional vertex 'normal' vector, $\boldsymbol{q}_1, \boldsymbol{q}_2, ..., \boldsymbol{q}_N$, to calculate the correct sign of the distance. This vector, is defined as the average of the normal unitary vectors of the adjacent boundary edges, i.e. the vertex 'normal', $\boldsymbol{q}_k$, of the vertex $(x'_k, y'_k)$, is calculated as:

$$\boldsymbol{q}_k = \frac{\boldsymbol{n}_k + \boldsymbol{n}_{k-1}}{2} \tag{4}$$

The vector $\boldsymbol{q}_k$ points to the 'correct' side of the vertex. This means that the sign of the projection $\boldsymbol{a}_{ik}$ on $\boldsymbol{q}_k$, $\sigma_{ik} = sign(\boldsymbol{a}_{ik} \cdot \boldsymbol{q}_k)$, is positive if the turbine is inside an inclusion zone and outside an exclusion zone, and vice-versa if the projection is



negative. To summarize, the signed distances $D_{ik}$ from all wind turbines $i$ to all boundary edges $k$ of all inclusion and exclusion zone polygons are calculated as:

$$D_{ik} = \begin{cases} |\boldsymbol{a}_{ik}|\,\sigma_{ik}, & \text{if } \tilde{a}_{ik} < 0 \\ \hat{a}_{ik}, & \text{if } \tilde{a}_{ik} \geq 0 \text{ and } \tilde{a}_{ik} \leq |\boldsymbol{e}_k| \\ |\boldsymbol{b}_{ik}|\,\sigma_{ik}, & \text{if } \tilde{a}_{ik} > |\boldsymbol{e}_k| \end{cases} \tag{5}$$

Note, in the second case in Eq. 5, the sign is implicit in $\hat{a}_{ik}$.

Hereafter, we can proceed with the next step, which consists of identifying the nearest boundary edge (of the nearest polygon). This is done by:

$$C_i = \min_k(|D_{ik}|) \tag{6}$$

which calculates a vector $\boldsymbol{C}$ whose components represent the distance between each turbine and their respective nearest bound-
ary edge. This vector is the term in the inequality constraint in Eq. 1. The gradient of $\boldsymbol{C}$ will define the right direction to go for every wind turbine to be placed within an inclusion zone and outside exclusion zones, to meet the boundary constraint. The analytical gradients of $\boldsymbol{C}$ with respect to the wind turbine coordinates can be written as follows:

$$\frac{\partial C_i}{\partial x_i} = \begin{cases} \dfrac{x_i - x'_k}{\sqrt{(x'_k - x_i)^2 + (y'_k - y_i)^2}} & \text{if } \tilde{a}_{ik} < 0 \\[3ex] \dfrac{y'_k - y'_{k+1}}{\sqrt{(x'_{k+1} - x'_k)^2 + (y'_{k+1} - y'_k)^2}} & \text{if } \tilde{a}_{ik} \geq 0 \text{ and } \tilde{a}_{ik} \leq |\boldsymbol{e}_k| \\[3ex] \dfrac{x_i - x'_{k+1}}{\sqrt{(x'_{k+1} - x_i)^2 + (y'_{k+1} - y_i)^2}} & \text{if } \tilde{a}_{ik} > |\boldsymbol{e}_k| \end{cases} \tag{7}$$

$$\frac{\partial C_i}{\partial y_i} = \begin{cases} \dfrac{y_i - y'_k}{\sqrt{(y'_k - y_0)^2 + (y'_k - y_i)^2}} & \text{if } \tilde{a}_{ik} < 0 \\[3ex] \dfrac{x'_{k+1} - x'_k}{\sqrt{(x'_{k+1} - x'_k)^2 + (y'_{k+1} - y'_k)^2}} & \text{if } \tilde{a}_{ik} \geq 0 \text{ and } \tilde{a}_{ik} \leq |\boldsymbol{e}_k| \\[3ex] \dfrac{y_i - y'_{k+1}}{\sqrt{(x'_{k+1} - x_i)^2 + (y'_{k+1} - y_i)^2}} & \text{if } \tilde{a}_{ik} > |\boldsymbol{e}_k| \end{cases} \tag{8}$$

Note that $\frac{\partial C_i}{\partial x_j} = 0$ when $i \neq j$, and that, in Eqs. 7 and 8, $k$ is set as $\operatorname*{argmin}_k(|D_{ik}|)$.



In general, the distances between the wind turbines and the boundaries are continuous and differentiable, but not continuously differentiable. When the nearest edge with respect to a wind turbine switches from one edge to another, the gradient will make a discontinuous jump, but this does not seem to be an issue for the used solver (see details of the solver in Section 3).

The gradient-based solver Sequential Least Squares Programming described by Kraft (1988), from now on SLSQP, was used for the numerical experiments, as implemented in the Python library *Scipy* (Virtanen et al., 2020). The running time of the algorithm scales linearly with both the number of wind turbines and the number of boundary edges ($N_{WT} \cdot N_{edges}$).

## 2.2 Distance relaxation

The described methodology to include multiple polygons in our optimization domain means that solvers, which seek to obtain a feasible solution immediately after the optimization is started, will try to satisfy the inequality constraint, $C_i \geq 0$, by placing all the wind turbines inside the nearest polygon. Moreover, once a wind turbine is inside a polygon, it will not be able to explore the rest of the design space as it would violate the boundary constraint, and therefore the amount of wind turbines allocated in each polygon will depend on the initial positions.

A potential solution for this is to relax the boundary constraints for a certain number of iterations. This allows for a deeper exploration of the design space at the beginning of the optimization, and thus a more suitable distribution of the wind turbines between the existing polygons. The relaxation is applied as a linear offset that is added up to the default distances and depends on the number of iterations. The relaxation is applied with a linear expression that describes the relation between the relaxed distances, $R_{ik}$, and the iteration number, $\gamma$:

$$
R_{ik} = \begin{cases} D_{ik} + k_r \left( \gamma_r - \gamma \right), & \text{if } \gamma < \gamma_r \\ D_{ik}, & \text{if } \gamma \geq \gamma_r \end{cases}
\tag{9}
$$

where the first parameter, $k_r$, defines the added offset per iteration, and the second parameter, $\gamma_r$, defines the number of iterations during which the relaxation is applied. With this implementation, the largest offset, $k_r \cdot \gamma_r$ occurs at the beginning of the optimization and is gradually reduced until the maximum number of iterations for relaxation, $\gamma_r$, is reached.

For a better understanding of the distance relaxation, its application has been illustrated in Figure 3. In the example, a simple set of polygons representing five inclusion zones is relaxed applying an offset of 5m per iteration ($k_r$=5), during 160 iterations ($\gamma_r$=160). The iteration number is given by $\gamma$, illustrated in the title of each plot. In the first iteration ($\gamma = 1$), the boundaries are relaxed with the maximum offset, which is beyond the limits of our domain, thus the relaxed polygons cannot be seen in the plot. The design variables are only limited by their respective upper and lower bounds, meaning that the solver can freely move them within the whole domain, illustrated by the red dashed line. When $\gamma = 100$, the relaxed polygons start to gradually push the wind turbines towards the inclusion zones as the optimization progresses. The plots for $\gamma = 130$, 140 and 150 illustrate this process in which the relaxed boundaries are shrunk until their default shapes. Eventually, for $\gamma = 160$, the relaxation ends and all wind turbines will be allocated inside a polygon.

When distance relaxation is applied, the nearest boundary is determined as:





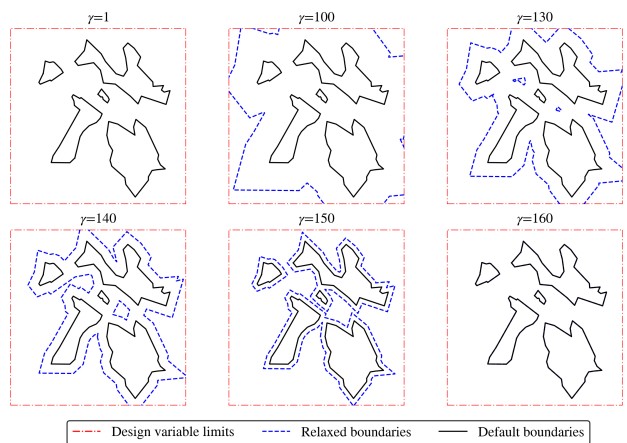

**Figure 3.** The optimization boundary constraint is visualized for different optimization iterations when using the distance relaxation method in a hypothetical problem with $k_r = 5$ and $\gamma_r = 160$. The solid black lines denote inclusion zone boundaries; the dashed red lines show the domain limits, defined by the upper and lower bounds of the design variables; the blue dashed lines represent the relaxed boundaries, which change depending on the iteration number $\gamma$. Each plot shows a different optimization iteration number.

$$C_i = \min_k \left( |R_{ik}| \right), \tag{10}$$

which is a vector of the nearest distances between each turbine and the relaxed boundaries.

The distance relaxation allows for a better exploration of the design space. It can also be applied to escape from local
optimum or to allow transferring of wind turbines between polygons for a certain number of iterations.

### 2.3  Smart-start algorithm

Another method to solve the wind turbine allocation problem is to discretize the domain and place the wind turbines in the inclusion zone polygons before the optimization is launched. The *smart-start* algorithm, implemented in PyWake as in Rodrigues et al. (2023 forthcoming), is meant to get a better initial layout of a wind farm. A diagram depicting the rationale of
the algorithm is presented in Figure 4. The idea is to sequentially place the wind turbines one by one in the positions with the best wind resource, taking wake effects of the previously added wind turbines as well as boundary and spacing constraints into account.

The algorithm takes a list of discretized potential wind turbine locations, $\mathcal{L}$, as input, and after removing all locations where the boundary constraints are violated, the main loop starts. In each iteration the wind resource including wake effects from
already added wind turbines is evaluated at all points in $\mathcal{L}$, and the next wind turbine is added at the location that yields the highest AEP. This means that the reduction in AEP of the previously added turbines is not considered. Hence the solution may





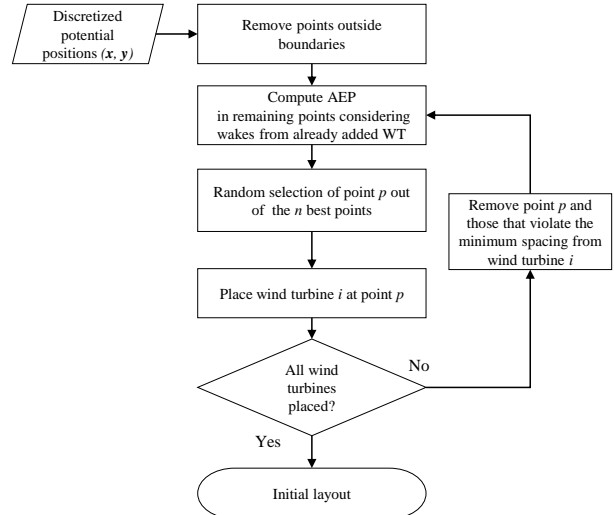

**Figure 4.** Workflow visualization of the *smart-start* algorithm.

not be optimal, but it implies a crucial reduction in computational costs. After adding the next wind turbine, all points where the spacing constraints are not satisfied are removed from $\mathcal{L}$, and the algorithm continues until all wind turbines are added.

The algorithm has been extended with a randomness parameter $r$ that allows it to put the next turbine at one of the $n$ best
positions by random, where $n = \max(r \cdot size(\mathcal{L}), 1)$.

The $r$ parameter ranges between 0, which corresponds to always picking the best position, and 1, which corresponds to completely random choice of points.

The performance of *smart-start* depends on the number of potential locations (grid resolution): A very high resolution grid will involve high computational effort, while a coarse resolution might lead to inefficient wind turbine allocation.

## 2.4   Flow modeling

When a site has complex terrain, like mountains and valleys, the landscape creates changes in pressure that result in changes in the local wind speed. This section explains how the wind resource and the flow are modeled along the study case. The wind farm is modeled using a flow map describing the local wind conditions. The local directions and speeds are dependent on the spatial location, the freestream wind speed, $U_\infty$, the freestream wind direction, $\theta_\infty$, and local velocity deficits from wake
losses. The local wind speed and direction can be expressed as:

$$U_{iud} = U_{\infty u} \cdot s_{id} \tag{11}$$

$$\theta_{id} = \theta_{\infty d} + t_{id} \tag{12}$$





where the terms $s_{id}$ and $t_{id}$ represent the wind speedups and turning respectively, at a location $i$ and for a freestream direction $d$, $U_{iud}$ represents the local wind speed, affected by the orography speedup effects, and $\theta_{id}$ represents the local wind direc-

tion, affected by the orography turning effects. The local wind speed has to include the deficits derived from wind turbine interactions, $\Delta u_{iud}$, which are imposed as:

$$u_{iud} = U_{iud} - \Delta u_{iud} \tag{13}$$

In this study, the objective function of the optimization is the Annual Energy Production or AEP. For each set of inflow conditions, the individual turbine powers are summed according to the probability of $(U_{\infty u}, \theta_{\infty d})$,

$AEP(\boldsymbol{x}, \boldsymbol{y}) =$

$$8760 \sum_{d=1}^{N_\theta} \sum_{u=1}^{N_u} \sum_{i=1}^{N_{wt}} P(u_{iud}) \rho(U_{\infty u}, \theta_{\infty d}) \tag{14}$$

where $\rho$ is the probability mass function, $N_{wt}$ is the number of wind turbines, $P$ is the power curve function, $u_{iud}$ is the local velocity, including wake effects, associated with freestream direction $\theta_{\infty d}$, wind speed $\mathcal{U}_{\infty u}$, and turbine position $(x_i, y_i)$.

The wake effects are approximated using the Bastankhah Gaussian wake deficit model (Bastankhah and Porté-Agel, 2014).

This model is derived from the mass and momentum conservation and assumes a Gaussian distribution of the velocity deficits in the wake, controlled by a single parameter $k^*$ to model the expansion. The velocity deficit from wind turbine $j$ on wind turbine $i$ is estimated by the expression below:

$\Delta u_{ijud} =$

$$U_{iud} \left( 1 - \sqrt{1 - \frac{C_\mathrm{T}}{8 \left( k^* \Delta_{ijd}^x / d_0 + \varepsilon \right)^2}} \right) \tag{15}$$

$$\exp \left\{ \frac{-1}{2 \left[ k^* \Delta_{ijd}^x / d_0 + \varepsilon \right]^2} \left[ \left( \frac{\Delta z_{ij}^H}{d_0} \right)^2 + \left( \frac{\Delta_{ijd}^y}{d_0} \right)^2 \right] \right\}$$

where $k^*$ is the expansion parameter (for this study, a value of 0.032 was used), $d_0$ is the rotor diameter of the wind turbines,

$\Delta_{ijd}^x$ and $\Delta_{ijd}^y$ are the upstream and crosswind distances between turbines $i$ and $j$ respectively, $\Delta z_{ij}^H$ is the hub height difference between turbines $i$ and $j$, $C_T$ is the thrust coefficient, and $\varepsilon$ is the standard deviation of the Gaussian profile, normalized with the rotor diameter, very close to the upstream wind turbine, i.e., where $\Delta_{ijd}^x \approx 0$.

In this study, $\Delta_{ijd}^x$ is computed as the length of the line that follows the terrain, i.e. has a constant height above the ground, from the upstream to the downstream wind turbine projected onto the downwind direction axis, $\Delta_{ijd}^y$ is the straight crosswind

distance, and $\Delta z_{ij}^H$ is zero, as all the wind turbines have the same hub height.





The resulting velocity deficit fields are calculated by adding up the deficits from the upstream wind turbines using a squared sum wake superposition model,

$$\Delta u_{iud} = \sqrt{\sum_{\forall j:\, \Delta^x_{ijd} > 0} \Delta u^2_{ijud}} \tag{16}$$

where the notation $\Delta^x_{ijd} > 0$ ensures that the velocity deficits are only accounted for downstream distances.

The wind turbine model used for this work is based on the Vestas V80-2.0, which has a rotor diameter of 80 meters, a hub height of 70 meters and a nominal power of 2 MW. The power and thrust curves for this model are pre-defined in PyWake.

## 3   Application

In this section, we present a case study based on a fictitious site, Parque Ficticio, with complex terrain. The site is featured by a non-uniform wind resource and a terrain elevation map. The purpose of the study is to show a wind farm layout optimization

using gradient-based methods and demonstrating the performance of the multi-polygon boundary constraint described in the prior section of this work. The results from the three different optimization approaches are compared and assessed.

### 3.1   Optimization setup

A wind farm consisting of 12 wind turbines is optimized following the formulation described in Eq. 1, where the design variables are the wind turbine locations, and the constraints comprise of a minimum spacing of 2 rotor diameters (equivalent to

160 m) between wind turbines and 5 irregular inclusion zones polygons (see Figure 5). The $AEP$ is calculated as in Eq. 2.4, considering local wind directions and speeds, as defined by Eqs. 12 and 13.

### 3.2   Site description

Parque Ficticio is a fictitious wind farm site predefined in PyWake, where the wind resource and terrain data are given as a dataset whose coordinates are $x$ (UTM easting projection), $y$ (UTM northing projection), $h$ (height) and $wd$ (wind direction sector). The wind resource is characterized by a unique Weibull ditribution per wind sector (12 sectors). The dataset contains

a gridded map of speedups and turning values that change with the sector.

At the site, we have defined 5 potential inclusion zones polygons as seen in Figure 5, where the wind turbines are allowed to be installed. From the wind rose in Figure 5, it can be observed that the site has dominant westerly winds. The local winds are affected by the terrain effects, such as speedups and turning.

### 3.3   Approaches to optimization of the layout


The wind farm is optimized following three independent approaches. The initial position of the wind turbines are randomized using predetermined random number generator seeds to ensure that the results are reproducible. Fifty seeds (from 1 to 50)





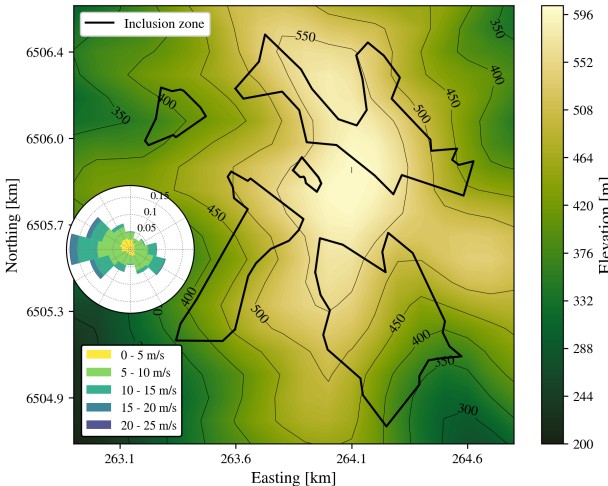

**Figure 5.** Parque Ficticio site is considered in this study. The elevation is shown with colors in contours. The inclusion zones are shown in thick dark lines. The illustrated wind rose shows the sector and wind speed frequency according to the freestream probabilities, which change locally due to orographic effects.

were run for each of the three approaches. For all approaches, the gradient-based solver SLSQP was used. The discretization of the wind directions and wind speeds for the optimizations were 1 degree and 1 m/s bins respectively. We set a limit of 500
iterations with a tolerance for convergence of $10^{-6}$ on the objective function.

The optimizations were run in a HPC cluster (Technical University of Denmark, 2019) to allow parallelization. Every case was performed in an individual node composed by 2x16 processors AMD EPYC 7351, @2.9 GHz, and a RAM of 128 GB. The numerical computations were parallelized using the full node capacity. A new Anaconda environment (ana, 2020) was created with the required Python libraries (see Section 5 for further information about Git repositories and used commits).
When performing AEP computations, PyWake allows "chunkification" of the wind directions and wind speeds to increase the computation speed. In our simulations, the wind direction flow cases were divided into 32 wind direction chunks.

The different approaches for optimization using gradient-based solvers and the implemented techniques are described below:

– **Approach 1** (SLSQP): the gradient-based solver SLSQP is used for the optimization. The Jacobians were calculated using automatic differentiation with the Python library *autograd* (Maclaurin et al., 2015). This same set-up is used for
the remaining approaches combined with other optimization techniques.

– **Approach 2** (Relaxation + SLSQP): the distance relaxation as described in Section 2.2 is applied during the first $\gamma_r$ iterations of the optimization. This allows the solver to freely move the wind turbines around the whole design space, leading to a better allocation of them between the different inclusion zones. The values of $k_r$ and $\gamma_r$ have to be selected accordingly to the size of the domain and the inclusion zone areas. A relaxation study was done to select suitable values
for the study case, see Section 3.4, resulting in $k_r$=100 and $\gamma_r$=100.





– **Approach 3** (*smart-start* + SLSQP): the *smart-start* algorithm is applied to achieve a better initial layout. This involves that all the initial positions are inside the inclusion zones at the beginning of the optimization. A 10% of randomization is used, which means that the positions are subsequently selected randomly out of the best 10% of available points. For this randomization, the indicated seeds are applied. The domain is discretized with a grid of $100 \times 100$ points. After the *smart-start* is executed, SLSQP is used as in the other two approaches. No distance relaxation was applied for this case.


The idea behind using these different approaches is to (1) prove that the nearest distance method succeeds to place the wind turbines inside the inclusion zones in a wind farm layout optimization problem; (2) demonstrate how the distance relaxation is able to achieve higher quality solutions by avoiding local optimum caused by the discontinuity of the boundaries; (3) show the advantages of using the *smart-start* algorithm to initialize the wind farm layout optimization problem with multiple boundaries, as it efficiently allocates the wind turbines between the inclusion zones.


### 3.4 Relaxation study

As described in Section 2.2, $k_r$ defines the offset per iteration and $\gamma_r$ indicates the maximum number of iterations for relaxation. $k_r$ can also be seen as the 'speed' of the relaxation, while $\gamma_r$ can be seen as the 'duration' of the relaxation. If these parameters are too small, the relaxation will not be effective, as the extension of the allowed area is too small or there might not be enough time (iterations) to explore the domain. On the other hand, if one of these parameters is too large, the optimization may converge before the boundaries are back to their true shapes, involving the risk of reaching a constraint-violating solution.


A parametric study was done in order to find a suitable combination of values that would lead to the higher potential yield for the study case. Four combinations of values for $k_r$ and $\gamma_r$ were chosen: Combination 1, 2 and 3 compare the impact of different relaxation speeds for a fixed number of iterations before the last 300 meters are relaxed (from 300 meters on, the changing boundaries begin to push the wind turbines towards the true boundaries). These combinations satisfy the expression:


$$\gamma_r = \frac{300}{k_r} + 100, \tag{17}$$

which ensures 100 iterations of optimization before relaxation is applied to the last 300 meters. In other words, the solver has 100 iterations to distribute the wind turbines within a sufficiently large area to find positions where the wind resource is higher before the last part of the relaxation happens. The selected $k_r$ values were: 2; 5 and 100 (in the case of $k_r = 100$, $\gamma_r$ would be 97 to satisfy Eq. 17, but it was rounded up to 100).


Combination 3 and 4 compare the impact of different relaxation speeds for fixed relaxation time with the idea of exploring the impact of the relaxation maximum offset (i.e., a low value against a high value of $k_r$ keeping a constant $\gamma_r$). The selected values for the combination 4 were $k_r = 2$ and $\gamma_r = 100$, which was compared to combination 3 with $k_r = 100$ and $\gamma_r = 100$.

A total of 50 optimizations were run for every of the selected combinations. Some of the seeds were filtered as they resulted in constraint-violating solutions. More specifically, Table 1 shows the percentage of seeds that failed to reach constraint-satisfying solutions. In total, 17 out the 50 seeds failed in at least one of the combinations and were filtered out, thus the final sample for this relaxation study contains 33 seeds that run successfully for all the combination of relaxation parameters. Figure 6 shows


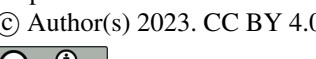



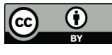

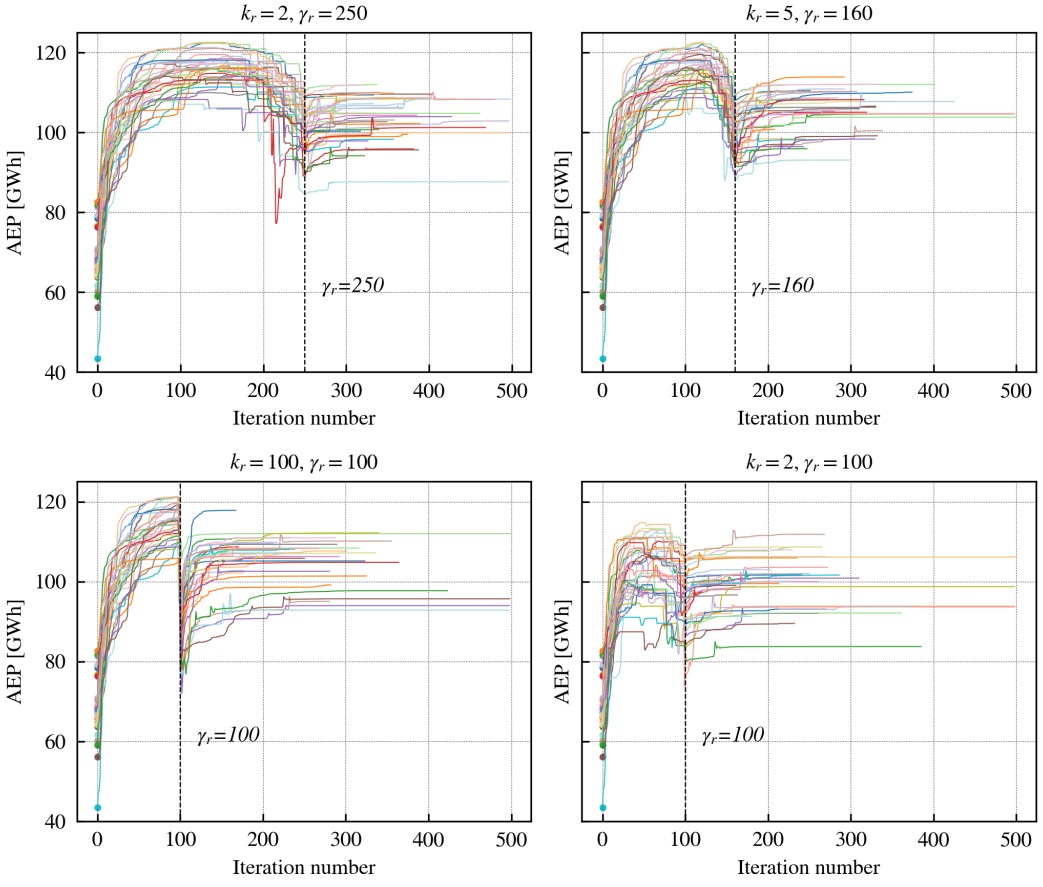

**Figure 6.** AEP convergence as a function of the iteration number for each of the studied relaxation combinations. The multicolored lines correspond to the different seed numbers.

the AEP of the 33 runs during the optimization as a function of iteration number. The dashed lines in each plot indicate the iteration where the relaxation finishes, and a different color is used for each seed. When the optimizations are launched, it can

be seen how the AEP increases quickly due to the effect of relaxation. When the wind turbines start to find positions that are close to local optimum, a plateau is formed for the curves (the plateau shape can be observed more clearly for the first plot, for $k_r = 2$ and $\gamma_r = 250$). Afterwards, when the relaxed boundaries get closer to the default inclusion zone polygons, the slope of the AEP curve becomes negative, as the turbines are forced to leave those already-found sweet spots and pushed inside the smaller allowed areas. Depending on the value of $k_r$, this slope is more or less steep: notice that for the lower left plot

($k_r = 100$ and $\gamma_r = 100$), this slope is almost vertical, as the last 300 meters of relaxation occurs in only 3 iterations (which would be very similar to removing the boundary constraints for a the first 100 iterations of the optimization and activating them afterwards).





**Table 1.** Failed seeds for the relaxation study.

| Relaxation strategy | Number of failed seeds | Percentage |
|---|---|---|
| $k_r = 2, \gamma_r = 250$ | 12 | 24% |
| $k_r = 5, \gamma_r = 160$ | 6 | 12% |
| $k_r = 100, \gamma_r = 100$ | 3 | 6% |
| $k_r = 2, \gamma_r = 100$ | 1 | 2% |

Figure 7 shows the statistics from this study using violin plots, where the mean and the first and third quantiles are repre-
sented. There are two remarkable facts that can be inferred from this plot: firstly, for the combinations 1-3 ($k_r/\gamma_r$ corresponding
to 2/250, 5/160 and 100/100) the mean of combination 3 with the fastest relaxation is slightly higher, but all distributions are
relatively similar. Based on this, we can state that there is no clear benefit in slowing down the relaxation, i.e., instantly activat-
ing the constraints after 100 iterations performs at least as good as a more gradually relaxation strategy. Secondly, from the last
two combinations ($k_r/\gamma_r$ corresponding to 100/100 and 2/100) we can conclude that the number of iterations before relaxing
the last 300 meters has an impact on the result, as for a too small $k_r$, the solver does not have time to find good position before
the relaxed boundaries are back to their default shapes.

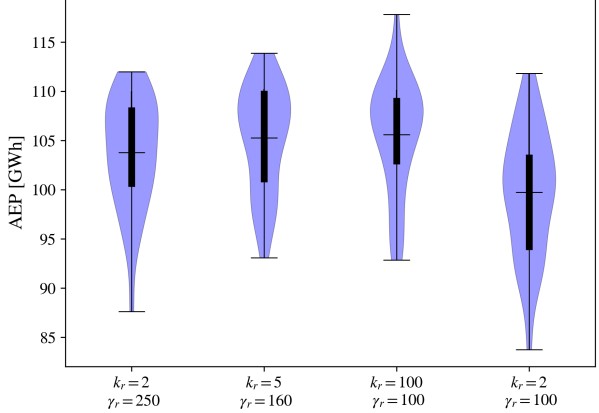

**Figure 7.** Relaxation study statistical summary. The violin plots illustrate the distribution of results from the 33 seeds. The black bars indicate
the first and the third quantiles of the sample. The horizontal lines represent the means.

Based on this relaxation parametric study, it was decided that $k_r = 100$ and $\gamma_r = 100$ would be used as parameters for the
relaxation approach, as this combination gives slightly higher AEP and fewer constraint-violating results.



## 4   Results

In this section, the results from the optimizations that were run following the described approaches are presented. Despite most
of the optimizations converged successfully, there were a few cases where they reached the maximum number of iterations
before convergence; these were considered as valid if all the wind turbines satisfied the boundary and the distance constraints.
On the other hand, some seeds led to an infeasible solution for some of the approaches; in these cases, the seed was not
considered for the final results. In total, 8 seeds had to be removed from the results because of this reason, as shown in Table
2. More specifically, the next events were the reason for discarding them:

1. SLSQP showed constraint violations after reaching the maximum number of iterations in seeds 9, 21, 22 and 33.

2. In the seeds 3 and 26, SLSQP + relaxation converges before the boundaries are back to their true shape, i.e., the op-
timization finishes in less iterations than the maximum number of iterations for relaxation, leading to an unfeasible
solution.

3. Approach 1 fails due to incompatibility of inequality constraints before reaching the limit of iterations in seed 29.

4. Approach 2 fails in seed 41 as SLSQP fails to find a constraint satisfying solution.

**Table 2.** Failed seeds for the different approaches.

| Optimization approach | Number of failed seeds | Percentage |
|---|---|---|
| SLSQP | 5 | 10% |
| Relaxation + SLSQP | 3 | 6% |
| *Smart-start* + SLSQP | 0 | 0% |

From the 42 remaining seeds, the first approach achieved an average AEP value of 95.55 GWh, with a standard deviation of
8.04 GWh. Approach 1 took an average of 207 iterations to finish and an average time of 152 seconds. The convergence of the
different seeds is illustrated in Figure 8.

With the second approach, an AEP average value of 105.3 GWh was yielded, with a standard deviation of 5.49 GWh. This
involves an increase of +10.2% of the AEP compared to the first approach. Approach 2 took an average of 303 iterations to
finish and an average time of 180 seconds. The convergence of the different seeds is illustrated in the center plot of Figure 8.
Until iteration 100, the AEP increases significantly, as the boundaries are relaxed and the solver can find wind turbine positions
in areas with higher resource; however, when the boundaries go back to their default shapes, some of the areas with good
resource become restricted and the solver has to seek for new ones inside the inclusion zones. The abrupt decay in the AEP
happens in a few iterations due to the high speed of the relaxation, as it was shown in the relaxation study (Section 3.4).

The third approach achieved an average AEP value of 115.17 GWh, with a standard deviation of 1.43 GWh. This involves
an increase of +20.53% with respect to the first approach, and a +9.37% with respect to the second approach. This approach

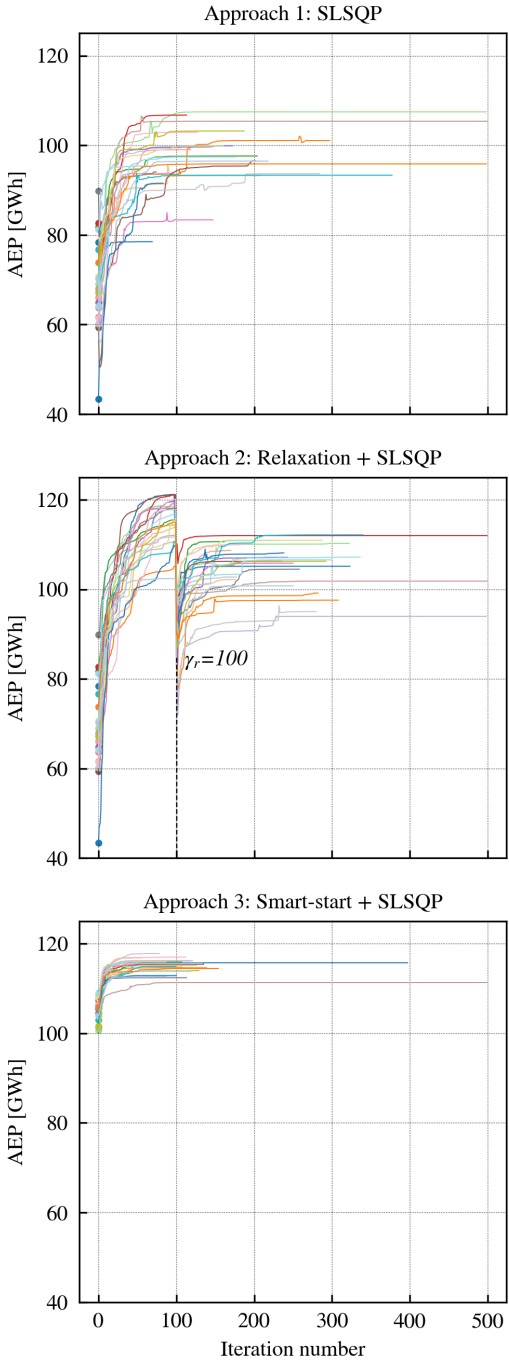

**Figure 8.** AEP plotted as a function of the iteration number of each optimization approach examined. The multicolored lines correspond to the different seed numbers.





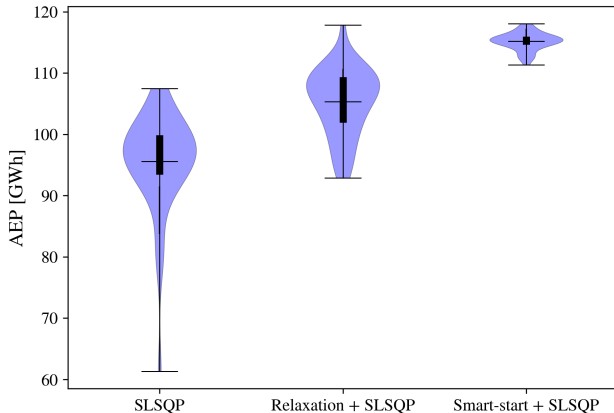

**Figure 9.** Violin-plots depicting the distribution of AEP found through the different optimization approaches using 42 random starting locations.

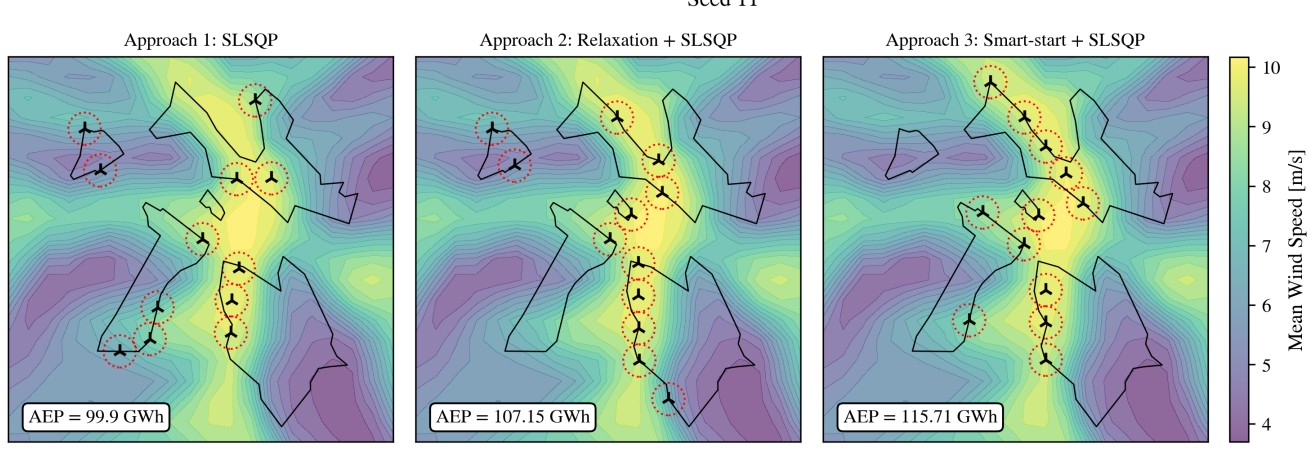

**Figure 10.** Final layouts, seed 11. The contour colors in the background indicate the mean wind speed over the site. The legend indicates the AEP for each of the resulting layouts for this seed. Approaches 2 and 3 succeed in placing more wind turbines over the yellow areas, where the mean wind speed is higher, leading to better solutions.

took an average of 149 iterations to finish and an average of 118 seconds, which includes the time required to compute the *smart-start* (on average 3 seconds using a regular grid of $100 \times 100$, which corresponds to spacing each point $2.375 \times 10^{-1}$
rotor diameters in each direction, $x$ and $y$). The faster convergence with respect to the other approaches is due to starting the optimization from feasible positions that already provide a good AEP, from where the gradients can find the local optimum easier.





Figure 9 shows a graphical summary of the statistics described above. The approaches subsequently achieve a higher AEP on average. The tail of the distribution for approach 1 is longer due to finding a local sub-optimum, but most of the seeds tend to find a optimum around the mean. The standard deviation decreases remarkably when using the *smart-start* algorithm, as many sub-optimal solutions are avoided. These results demonstrate that the technical limitation of the method can be overcome with the relaxation of boundaries, and moreover, the initialization of the layout with the *smart-start* provides better solutions than using random guesses all over the domain. On top of that, the *smart-start* converges faster towards the solution, requiring on average 27% to 35% less of time when compared to the other approaches.

Figure 10 illustrates an example of the final layouts achieved by one of the seeds (number 11), which is representative of an average result. The contours in the background represent the mean wind speed at the hub height. The locations with higher resource in the map correspond to the areas with higher elevation, and looking back to Figure 5, it can be recalled that the wind has majorly a westerly component, which explains why the wind turbines tend to align in the vertical direction. The legend in each of the plots indicate the total AEP achieved by the layout, and the dotted red circles represent the minimum spacing constraint.

At first glance, it can be noticed that the allocation of wind turbines is better when the relaxation or the *smart-start* are applied. In the layout achieved by approach 1 (Figure 10 plot in the left), the discontinuities between different inclusion zones increase the odds for the solver to become stuck in local minimum, which results in the baseline approach often failing to allocate the wind turbines in areas with higher resource. An instance of this is the four wind turbines in the lower left inclusion zone; despite of the wind resource being lower than in other areas, the solver opts to place them in that polygon as it was nearest to their initial positions, and during the optimization these wind turbines are not leaving the polygon as it would violate the boundary constraint unless they manage to jump into another inclusion zone.

The relaxation of boundaries skips this local optimal traps temporary and as a result achieves an improvement in the farm AEP (middle plot in Figure 10). It can be observed that in the same inclusion zone (lower left polygon) there is three less wind turbines than for approach 1. The result of this is a significantly higher yield, leading to an AEP of 107.15 GWh (+7.25% with respect to approach 1).

The *smart-start* beats boundary discontinuity in a different way: the wind turbines are placed one by one in the positions with best resource, leading to a initial feasible solution that, despite not allowing wind turbine transferring between the inclusion zones, it provides an already good distribution of them within the available polygons. In this particular case, it can be seen how the approach 3 (right plot in Figure 10) allocates more wind turbines where the yield potential is higher.

In general, the use of the *smart-start* to find a better initial layout before the optimization proved to achieve higher quality solutions; the relaxation applied during a number of iterations when the optimization is started helps to get higher values of AEP if compared to the use of SLSQP alone.





# 5 Conclusions and future work

This article describes a new vectorized method to include multiple boundary constraints represented by polygons in wind farm layout optimization problems. The method relies on the nearest distance from the wind turbine positions to the polygons. The sign of the distance determines if the wind turbine is inside or outside of the polygon. Positive sign involves that the wind turbine is inside an inclusion zone or outside a exclusion zone, and vice-versa for the negative sign. A new inequality constraint is introduced in the optimization formulation to force the wind turbines to stay within the desired polygons.

Despite a limitation of this methodology being identified, which relates to the correct allocation of wind turbines between polygons, two potential solutions are proposed: the implemented boundary constraints can be relaxed during a definite number of iterations to allow a better exploration of the domain, and consequently finding a better allocation of turbines; alternatively, a heuristic algorithm can provide a better initial guess that saves computational time and achieves higher quality solutions.

To demonstrate the applicability of the method and the effectiveness of the proposed solutions to its limitation, a study case 425 was presented. A wind farm consisting of 12 wind turbines in a site with non-uniform wind resource and elevation was optimized using three different approaches: the approach 1 consisted on using a gradient-based solver, while the approaches 2 and 3 were subsequent combinations of the same solver with the boundary relaxation and the *smart-start* algorithms, respectively. For the numerical computations, the open source Python libraries developed at DTU Wind Energy, TOPFARM and PyWake, were used.

The results show that the distance method successfully achieves to respect the boundaries of the given irregularly shaped and disconnected polygons, although for certain initial conditions, incompatibility between the spacing and the boundary constraints might lead to an unfeasible solution. A parametric study for the relaxation was done with four combinations of parameters to decide the most suitable values that would be used in the study case. This demonstrated that the speed of relaxation does not have a significant impact on the results, but there has to be enough time for the solver to place the wind 435 turbines around the areas with higher resource. In general, the relaxation of boundaries proved to help finding a better optimum layout and wind turbine distribution between the inclusion zones. An average improvement of +10.2% in the AEP gain was achieved by the approach 2 with respect to sole use of gradients to solve the optimization problem. Moreover, the combination of heuristic optimization methods such as the *smart-start* algorithm, with gradient-based solvers followed by the approach 3 reached even higher quality solutions while saving some computational time. The average improvement of the approach 3 with 440 respect to the other were +20.53% and 9.37% respectively.

In future work, the method should be tested under more realistic scenarios, with higher number of inclusion zone polygons and wind turbines, which increases the computational cost substantially. We observe that the time used by SLSQP to handle constraints becomes significantly around 10,000 constraints. Furthermore, the comparison of the described approaches could be investigated using different solvers.



*Code and data availability.* The code used in this study is available on DTU Wind Energy Systems' Gitlab (DTU Wind Energy Systems, 2022a, b).

*Author contributions.* JCR prepared the first draft, devised the idea for the study case, contributed to the relaxation implementation in the code, led the numerical computations and realized the analysis of results; RVR contributed with the literature review, methodology and experiment design, and was involved in the numerical computations; JQ contributed to the literature review, problem formulation,
methodology and experiment design, and the drafting of the manuscript; MFM developed the methodology, implemented it in the software framework and contributed to the problem formulation; MMP implemented the *smart-start* and the automatic differentiation in the software, and contributed to editing the manuscript; PER contributed to the design experiment and the development of the methodology.

**Aknowledgements**

The authors gratefully acknowledge the computational and data resources provided on the Sophia HPC Cluster at the Technical
University of Denmark, DOI: 10.57940/FAFC-6M81.



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
