# Peer review of "Gradient-based Wind Farm Layout Optimization With Inclusion And Exclusion Zones"

_Wind Energy Science, 2023_

## Author Comment (AC1)

**Reply to the referee 2**

October 18, 2023

We would like to thank the referee for the feedback and suggestions to improve the article. The technical comments (or typos) have been corrected directly in the manuscript, while the questions that require a more detailed explanation have been answered in this document, also modifying accordingly the respective part in the manuscript. Besides, some of the comments were already addressed since they had been already pointed out by the referee 1. We encourage the referee 2 to take a look at the response to the previous referee if the answer provided here seems too short. The answers are indicated in blue throughout the whole document.

1. Why are approaches using polygons or penalty functions not sufficient, in the opinion of the authors? A few details would be welcome here.

   In the literature review, we observed that commonly the concave shapes involve a challenge when using polygons, and this is something that our formulation can deal with in a robust way. Regarding the penalty functions, as we also mention in our answer to the referee 1, we are aware that it could be applied to the problem, but we just simply decided to go for this alternative. Besides, as also mentioned in our reply, applying a penalty function would require some type of smoothing term to avoid turbines oscillating in and out of the exclusion zones.

2. "If more than one polygon of the same type overlap, these are merged into a single one". This is fine, in principle. But what about inclusion and exclusion zones - I assume that these are not allowed to overlap?

   Both types are allowed to overlap. This is a very nice feature of the method, as we can first define all the polygons, assigning the inclusion or exclusion attribute to each of them, and then superpose all of them. The resulting shapes are the merged shapes created by the interception of all shapes. If two shapes do not intercept, but overlap, then depending on the case there are two options. To illustrate this let's imagine a big circle area and a second circle with the same center but smaller radius. If the larger circle is an inclusion zone and the smaller circle is an exclusion zone, then the resulting inclusion zone from the merge will be a ring; in the opposite case, the resulting inclusion zone from the merge will be the smaller circle.

3. What is a "unitary" vector? I assume the authors mean that these vectors are normalized to unit length?

   That is correct. We have added the next line to the manuscript in order to make it more clear: "(...) we define a normal unitary vector (a vector whose module is equal to the unit of length) that points inside the inclusion zone polygons and outside the exclusion zone polygons (...)".

4. It is very confusing to see the normal vectors drawn at the starting node of their respective edge, instead of - as I would expect - somewhere around the middle of the edge. Mathematically, there is no problem, but it goes somehow against expectations of how to visualize these things. Please reconsider.

   This is a fair point. We have corrected the Figure in the manuscript with the normal vectors drawn in the middle of the edges.

5. I would assume that this "greedy" algorithm could also fail under certain conditions? Moreover, since it seems to be deterministic (when r = 0), there would then be no starting condition for the actual optimization? Is this the (main) reason for adding randomness to this procedure? (I think there are

straightforward ways to deal with this problem better, though - maybe something the authors might want to think about)

This question requires an answer of several parts. For the first part: yes, the algorithm can fail under certain conditions, and one of those conditions is that there are not enough points to satisfy the constraints. For example, let's imagine a square with 5 rotor diameters of side length. If we want to allocate 4 wind turbines with a spacing constraint of 5 rotor diameters between each turbine, they would only fit in the square if they are located on the corners. Therefore, there is a feasible solution but the algorithm would fail if it places one of the turbines in a different position than a corner.

About the algorithm being deterministic, we need to clarify that it would be deterministic for this case, without using randomness, because the wind resource is not uniform, but it won't be deterministic for a uniform resource site since the first wind turbine will be always random and that would affect the final result. The randomness might still be useful with a non-uniform resource in certain cases, for instance, let's imagine a case with 2 polygons and non-uniform wind resource. One of this two polygons has much more resource than the other, but very limited space, whereas the other polygon has plenty of space to locate turbines. Now let's also imagine that the polygon with higher potential has a shape that, due to minimum spacing constraints between turbines, if one of them is put in the middle, then there won't be space for additional turbines within, whereas if we put the turbine in one of the corners, there will be space for another turbine. In a case like this, placing two wind turbines in the polygon with higher resource would lead to higher AEP than placing only one of them, and if the middle of the polygon would be slightly better, the smart-start with 0 randomness would place a turbine in the middle, not reaching the best solution.

6. Eq. 15: I assume the gradients of the velocity deficit are also needed, when determining the gradient of the AEP? But this is handled by algorithmic differentiation, correct?

Yes, we use the python library autograd to differentiate the code automatically, so we did not include this expression in the manuscript.

7. line 314ff: "there might not be enough time (iterations) to explore the domain". Well, the relaxation of the distance by way of Eq. 9 prescribes a specific way in which this constraint is changing from iteration to iteration. In some cases the optimization even terminates when the distance function is still relaxed (line 361), which the authors consider to be invalid optimization runs!? The first seems somewhat ad-hoc, and the second seems somewhat strange. I would imagine that it would be much more natural to fully optimize the wind farm layout for a fixed offset of the distance, and only then adjust the constraint and solve the next (slightly less relaxed) optimization problem - using the previous optimal solution as an initial (though most likely infeasible) state. Thus, instead of considering one optimization problem with continuously changing constraint relaxation this would be a series of optimization problems with different (but fixed) constraint relaxations - until the final problem with no relaxation is solved. Why did the authors not try this, very often used, strategy?

The only reason why we discard the optimization runs in which the solver terminates before the relaxation finishes (or better said, before the un-relaxation finishes), is due to the fact that some of the wind turbines might find the sweet spot out of the true boundaries. That means that this solution is unfeasible. The approach described by the referee (allow for the optimization to find the optimum and then restart it with smaller bounds) makes total sense, but it would take more time, and hence we would lose one of the desired features to use gradient-based optimization: the speed.

8. Continuing from the previous comment, there might also be better ways to change the relaxation offset to the distance adaptively, i.e., depending on the speed of convergence (how much better the solution becomes between iterations).

We agree that the relaxation function is somehow very basic and could be improved. Initially, the scope of the paper was to demonstrate the method to include polygons as boundaries while using the gradient-based method, so we did not explore very much the posibilities that could be applied both to the relaxation function or to the smart-start algorithm. This is rather something that could be considered for future work.

9. line 321: "before relaxation is applied to the last 300 meters." - There seems to be some confusion about the use of the term "relaxation" and I would like the authors to review their use of it. The relaxed optimization problem has the least strict constraint. It is thus not correct to talk about relaxing the constraint in later iterations where the constraint actually becomes more strict! Similar issues of imprecise use occur in line 346 ("no clear benefit in slowing down the relaxation") and line 375 ("high speed of the relaxation").

   This point was also made by the first referee, and we adapted the manuscript to his suggestion, changing the term 'relaxation' by 'un-relaxation' when it proceeds. We totally agree with this and we hope that the main text now has become more understandable for the reader.

10. line 325: Why the value of 97? Why not 103? There seems to be some inconsistency between this here and Eq. 17.

    This was a numerical error. It has been corrected in the main text.

11. Table 2: A comment seems appropriate why not both relaxation and smart-start have been tested together.

    The combination of the relaxation technique and the *smart-start* would place the turbines within infeasible areas and then, in many cases, the optimization would converge so fast that we would end in one of the situations in which we discarded those seeds. In any case, we do not believe that it would get better solutions that the single use of the *smart-start* combined with SLSQP. We did not explain this in the main manuscript because we think it might lead to confusion.

12. page 17. The results are discussed mainly in terms of average AEP, but what about the maximum values? These are, after all, what one is looking for when trying to optimize the problem!

    That is a fair point. Since the article is focused on the methodology, we decided to analyze the results from the statistical perspective, but in practice, it seems more reasonable to focus on the solutions that lead to highest AEP. However, it can be seen from the violin plots that the maximum solutions reached by the relaxation approach and the *smart-start* are very close.

13. Fig. 9: I am somewhat surprised to see so much variability in the results, depending on the used random seed / initial position of the turbines. Obviously there seem to be many local maxima in this problem. Although the approach of the authors is promising, this seems still to be a major problem - how to find the actual, global optimum? Some thoughts along this direction would be highly appreciated in the discussion.

    Indeed, it is a multimodal optimization problem with a lot of constraints that is solved with a gradient-based solver. This involves that the found optimum will be sensitive to the initial guess. On top of that, the combination of wind turbines per polygon also plays an important role in finding the global optimum. Unfortunately, it would be challenging to ensure that the global optimum has been found, as we are working on the continuous space and it becomes impractical to exhaustively explore all the possibilities.

14. Coming back to the problem of local maxima, have the authors considered to use the idea of marginal improvements - i.e., estimating numerically how much the addition or removal of a single wind turbine from each inclusion zone would add / remove from the overall AEP (per turbine)? This type of sensitivity could then be used to determine when or how to move turbines between inclusion zones...

    This approach is very interesting and we did not considered it. When thinking about how to put it into practice, there is the additional problem of deciding where to put the added / removed wind turbine, and given that the domain has variable wind resource, it becomes an optimization problem within another optimization problem (nested problem). We imagine this as a time consuming problem but will be kept in mind for future development.

---

## Author Comment (AC2)

**Reply to the referee 1**

October 18, 2023

We would like to thank the referee for the feedback and suggestions to improve the article. The technical comments (or typos) have been corrected directly in the manuscript, while the questions that require a more detailed explanation have been answered in this document, also modifying accordingly the respective part in the manuscript. The questions have been separated according to "major comments" and "minor comments", as received in the feedback. The answers are indicated in blue throughout the whole document.

**Response to major comments**

1. Penalty methods: The introduction describes how penalty methods have been used to handle boundary constraints in other papers, but the downside is the tunable hyperparameter. Approach 1 for the optimization uses no penalty methods, and the major downside is that the turbines are fixed in their respective inclusion zones after their initial placement. If a penalty parameter were included in Approach 1, wouldn't the turbines be able to jump between the inclusion zones more easily (as discussed around Line 402), and the algorithm would perform better ? I suppose that the convergence speed would suffer as a result, but I would expect the solutions to compare better to Approaches 2 and 3.

   We decided to use Approach 1 as a reference because it is what can be found in open source Python libraries (Scipy, Virtanen et al. (2020)), and therefore what everybody had access to. The idea was to depart from what we were already capable of doing in Topfarm (DTU Wind Energy Systems (2022); Larsen and Réthoré (2013)), and demonstrate how we designed a solution to the wind turbine allocation problem. There might be multiple ideas that would work in combination with SLSQP, for instance, a penalty method. Besides, gradient-based methods using Lagrange multipliers generally perform better than penalty methods (e.g., Weyler et al. (2012)). Penalty methods would result in turbines being moved in and out of exclusion zones or regions of nearby turbines. A penalty method would require some sort of smoothing, which the Lagrange multiplier provides, avoiding these oscillations. In the article we present the relaxation strategy, which was conceived to provide an solution for an industrial partner. Future work could explore more ideas to improve the quality of the solutions and the speed of convergence.

2. Boundary relaxation: I find Section 3.4 confusing to digest. Starting at Line 317, what is the importance of the "last 300 m?" Does 300 m force the inclusion zones to combine with one another, similar to the first panel in Fig. 3? Second, does the un-relaxation at the rate $k_r$ start at 100 iterations and continue until the limit $\gamma_r$ is reached? If so, how does Eq. 17 reconcile with Eq. 9? Or is the return to the actual boundaries being done continuously at the rate $k_r$ starting at the first iteration? Lastly, on Line 324: if $k_r = 100$, $\gamma_r$ should equal 103, not 97?

   We agree that this section might be described more accurately. On the first place, the last 300 meters of un-relaxation were chosen arbitrarily to focus on the part when the movement of the boundaries pushes the wind turbines inside the inclusion zones. The polygons are still merged as a whole when 200 meters of relaxation remain, as can be seen in the Figure 1, but the last 300 meters cover almost all the areas of the domain, confined by the design variable limits.

[Figure]

Figure 1: Relaxation: last 300 meters vs last 200 meters.

Secondly, Eqs. 9 and 17 are totally independent from each other. Eq. 9 describes how the relaxation strategy works, whilst Eq. 17 is just an expression to approximate the relaxation parameters effectively for this particular case, and also for the purpose of the demonstration. That said, the return to the actual boundaries is done linearly and in a continuous way at the same rate, $k_r$.

Lastly, on Line 324, that was mistake. Indeed, it should be 103 and not 97. It has been corrected in the manuscript accordingly.

3. Line 298 mentions how "The values of $k_r$ and $\gamma_r$ have to be selected accordingly to the size of the domain and the inclusion zone areas." This sentence is vague to me: for a larger domain with more turbines, would the number of iterations increase? If the inclusion zone areas are more complex, would these values increase?

Unfortunately, there is not a rule of thumb for this. A larger domain with more wind turbines will not necessarily require a larger total offset, for instance, if the inclusion zones were close to each other, but would probably benefit from a slower un-relaxation. At the same time, there might be a case with few wind turbines and very spread polygons, which would require a large total offset for the relaxation to be effective. The complexity of the inclusion zones shape is not as relevant as the number of zones and how spread they are.

4. Are there heuristics for how many iterations the optimization will take such that $\gamma_r$ is chosen to be smaller than that expected limit?

We are not aware/ignore the existence of any heuristics of that type.

5. Also, I think calling $k_r$ a relaxation speed (Line 319) is confusing because it's really an un-relaxation speed . There's surely a better name than that but 'relaxation' implies that the relaxation is progressing with $k_r$, not being undone (i.e. boundaries are returning from their relaxed state to the normal state).

We fully agree with this. The manuscript has been updated accordingly and now we speak about un-relaxation when we refer to the boundaries returning to their actual shapes.

6. Lastly, Line 347 discusses the differences between the performance of Combinations 3 and 4. The authors explain this by saying that Combination 4 does not have enough "time to find good positions", but I think this really has to do with the size of the relaxed inclusion areas. The number of relaxation iterations multiplied by the relaxation speed is the maximum offset of the boundary, and with the smaller $k_r$ and fixed $\gamma_r$, the relaxed boundaries are not sufficiently large to really explore the unconstrained solution space.

Fair point. It seems more reasonable to explain that the total or maximum offset is not sufficiently large rather than referring to the "time". We have corrected this in the manuscript.

**Response to minor comments**

1. Line 74: Can more detail be given on why gradient-based optimization would be ineffective? I suppose you're inferring that the high sensitivity to the initial conditions would require the user to run many optimization studies with randomized initial conditions to find the best candidates, which would be more inefficient than the proposed methods.

   The high sensitivity to the initial positions may lead always to bad solutions (independently from the number of run seeds). For instance, we can imagine a case with a very large inclusion zone is in the center of our domain, and let's assume that the wind resource is not very good within this large polygon. Then let's assume many other small inclusion zones located around, where the wind resource is richer. When the optimization is started, it is more likely that a wind turbine is allocated in the large polygon, while some of the small polygons could even remain empty. This is what we mean by inefficiency in this context.

2. Line 78: "This is achieved by adding an offset to the distance determined by the method." This sentence is vague —adding an offset to the distance to the boundary of the polygon?

   We have changed the description in the manuscript to make it more clear. The new text says as follows: *The first solution is to introduce a term in the boundary constraint formulation that relaxes the problem by expanding or buffering the inclusion zone areas before the optimization is started. Larger inclusion zones means that more of the domain can be explored and wind turbines can be placed around areas with better resource. During the optimization, the boundaries are un-relaxed linearly until they return to their true geometry. This is controlled with two parameters that model the offset per iteration and the number of optimization iterations in which the un-relaxation is applied.*

3. Eq 1: Maybe a preface about the lower and upper limits of the design variables could be included here. In Fig. 3, it is clear why these limits on the computational domain are necessary, but from Eq. 1 it appears that the function $C_i$ makes them redundant (since the reader assumes that the inclusion zone boundaries are within the computational domain).

   Fair point. We have added the next line in the manuscript: *Although it might seem redundant to include design variable limits due to the fact that the boundary constraint is already acting as such, they are used in further sections to describe the methodology.*

4. Eq. 4: Can more detail be given on why this step is necessary, as opposed to just using the sign of the distance to the nearest vertex ($a_{ik}$ or $b_{ik}$)? It seems that you'd just want to move towards that nearest point on the boundary. Also, where do the 'k' and 'k-1' indices come from? Everything else in this section is with respect to indices 'k' and 'k+1'.

   We need Eq. 4 because there might be a case like the one represented in Figure 2, where we observe an inclusion zone polygon with three boundary edges ($k$, $k+1$ and $k+2$) and two wind turbines (blue and red dots). The vectors $\boldsymbol{a}$ and $\boldsymbol{b}$ go from the vertex of their nearest boundary edge, which is the intersection between $k$ and $k+1$, til the wind turbine locations. If we follow the formulation of the method, Eq. 3 projects the vectors $\boldsymbol{a}$ and $\boldsymbol{b}$ on the direction of the normal unitary vectors of the nearest boundary edge. Since in this case we deal with a vertex, there are two potential nearest boundary edges: $k$ and $k+1$ (Figure 2, left). Therefore, following Eq. 3 we need to compute the projections of $\boldsymbol{a}$ and $\boldsymbol{b}$ on $\boldsymbol{n}_k$ and $\boldsymbol{n}_{k+1}$:

$$\hat{a} = \boldsymbol{a} \cdot \boldsymbol{n}_k$$
$$\hat{b} = \boldsymbol{b} \cdot \boldsymbol{n}_k$$

$$\hat{a} = \boldsymbol{a} \cdot \boldsymbol{n}_{k+1}$$
$$\hat{b} = \boldsymbol{b} \cdot \boldsymbol{n}_{k+1}$$

Since the wind turbine is outside the inclusion zone polygon, these projections should be negative. However, we see that in the case of $\boldsymbol{a}$ projected into $\boldsymbol{n}_{k+1}$, the result is positive. This means that if $k+1$ would be taken as the nearest boundary, the gradients would push the wind turbine away from the polygon. If we apply Eq. 4 (Figure 2, right), we can see how the projection is always negative, independently from the boundary edge that is used to project on.

$$\hat{a} = \boldsymbol{a} \cdot \boldsymbol{q}_k \leq 0$$

[Figure]

Figure 2: Left: vertical projections for the vectors $\boldsymbol{a}$ and $\boldsymbol{b}$ without Eq. 4. The projection on $k+1$ is positive ($\hat{\boldsymbol{a}}_{k+1} \leq 0$), leading to consider the wind turbine inside the inclusion zone, which is false. Right: vertex 'normal' vector $\boldsymbol{q}_k$ determines unequivocally the right sign for the projection of $\boldsymbol{a}$, independently from the boundary edge that is used for the calculation.

For the second part of the question, in the description of the method we use 'k' and 'k+1' to leave clear that it does not matter which vertex we use from the given polygon ('k' represents a generic vertex).

5. Eq. 6: A sign is needed here to denote which side of the boundary the turbine is on. Right now, the absolute value makes $C_i >= 0$ always, which would satisfy the constraint even when the turbine is outside of the inclusion zone.

Fair point. The code keeps the correct sign from the corresponding matrix component, but we did not reflect that in the formulation. We have adjusted Eq. 6 in the manuscript as follows:

$$C_i = \min_k(|D_{ik}|) \cdot sign(D_{ik}) \tag{1}$$

6. Eq. 11: The subscript 'u' should be defined similar to the subscript 'd'. Also, are the authors assuming that the speedup factor does not depend on the wind speed? That seems reasonable, but the assumption could be stated explicitly.

   We have fixed the definition in a way that now the subscript $u$ is also mentioned. Regarding the speedup (and turning), these values are defined in the site dataset as a function of the wind sector. This has also been described in the manuscript to clarify it.

7. Line 254: Why does the crosswind distance not follow the terrain like the upstream distance does?

   Currently the method does not consider the terrain variations in the crosswind direction. Future versions of PyWake will correct this.

8. Line 270: AEP is from Section 2.4, not Equation 2.4, and the wind speed and direction are defined in Eqs. 11 and 12, not 12 and 13.

   We wanted to refer to Eq. 13 rather than 11 because it includes the wake effects. We have changed the text to include all the equations: *The AEP is calculated as described in Section 2.4, considering local wind directions, local wind speeds and wake effects, as defined by Eqs. 11, 12 and 13.*

9. Line 275: 12 wind sectors are mentioned here, but then Line 284 mentions 1 degree wind direction resolution for the optimization. What is the meaning of the 12 wind sectors?

   The site has a resolution of 12 wind sectors (30° sector width). This means that the probability of combinations within the same sector will be equal. For instance, the probability of a wind speed of 10 m/s and a wind direction of 17° will be the same than the probability of a wind speed of 10 m/s and 28°, as 17° and 28° are within the same sector (15° - 45°). On the other hand, the bin resolution for the optimization is 1 degree, which means that wake losses will be optimized with 1 degree precision. We have added the next lines to the manuscript in order to clarify this aspect: *Notice that although the site has a resolution of 12 wind sectors, the bin resolution for the optimization is thinner, which means that wake losses will be optimized with 1 degree precision despite the frequency of certain inflow conditions is assumed to be the same.*

10. Line 290: Can this "chunkification" be explained more? Does this mean that the wind direction resolution is not 1 degree, or is this something to do with how the vectorized wind conditions are distributed across the computational nodes?

    We agree that this could be clarified. We have modified the corresponding part of the manuscript by writting: When performing AEP computations, PyWake allows "chunkification", which distributes the flow cases between the available resources in batches of wind directions and wind speeds, resulting in a reduction of the computational times.

11. Line 294: Autograd is used for automatic differentiation of the objective function and also the spacing constraint? I assume that the analytic gradients of the boundary constraint function (Eq. 7 and 8) have been provided to the optimizer.

    Autograd differentiates the code lines, and it is used both for the cost function and for all constraints (boundary and spacing). We don't need to provide additional information to the optimizer.

12. Line 326: I would be careful to clarify between optimization "time" and "iterations" throughout the paper. At some points, "time" is used where the authors are really referring to iterations and convergence.

    We have modified the parts where relaxation is described. Now we have tried to make clear that "un-relaxation" is referred to the part where boundaries change at a linear rate (towards their original shape). Also we have avoided the word "time" when we refer to the number of iterations for relaxation.

13. Line 329: Is there any explanation why these seeds fail beyond that the solutions violate the constraints? I suppose it's tied to the fact that the longer relaxation periods with slower relaxation speeds allow the optimizer to settle into local optima that are infeasible, but maybe that could be explained a bit more again.

    These seeds fail because the un-relaxation takes too many iterations, i.e., $\gamma_r$ is too large. In the Table 1 of the manuscript, it can be observed how the number of failed seeds increases as $\gamma_r$ is larger. This happens because the solver finds the optimal locations for the wind turbines before the design variables $x$ and $y$ are affected by the change of the boundary shapes that are being un-relaxed. If the distances are positive, the boundary constraint is satisfied and the solver assumes the optimum is found. We have changed the text to make it more clear: *Some of the seeds were filtered as they resulted in constraint-violating solutions, as a local minima is found before the un-relaxation finishes and the solver stops.*

14. Line 364: What does "incompatibility of inequality constraints" mean?

    This means that the solver is not able to place all the wind turbines within the inclusion zones while complying with the spacing constraint at the same time, and also it is not able to find a find new positions for the turbines that are violating the constraints.

[Figure]

Figure 3: AEP of the initial layouts as a function of the randomness.

15. Line 378: The lower cost of the smart-start optimizations, even including the time for the initialization (because the convergence is faster), is a neat result. Is there a sensitivity of the smart-start results to

the choice of the randomness parameter? How much randomness is required when initially placing the turbines to compensate for the unknown potential optimal AEP in each inclusion zone based on that number of turbines?

The random parameter has an impact on the results. In general, 0% randomness in a site with non-uniform wind resource will lead to the same layout independently from the seed number, as observed in Figure 3. In a site with uniform wind resource, the first wind turbine will be poistioned randomly anyway, and therefore we will observe certain variability on the results. Increasing the randomness will lead to higher standard deviation in the yield, which can also be observed in Figure 3. For this case, there is not a random value that leads to higher yield and compensates for the unknown potential optimal AEP.

16. Line 415: What is meant by "vectorized" here? Are the wind conditions computed in PyWake in a vectorized fashion? Are the constraints implemented in TOPFARM in a vectorized fashion? Vectorization has not been mentioned up until this point.

In PyWake, all the computations are vectorized. The methodology is also implemented in a vectorized way (distances to boundaries from every wind turbine), which makes it very fast. Nevertheless, we have removed the word from this line to avoid any confusion, given that it is not relevant for the article.

**References**

DTU Wind Energy Systems. Topfarm. `https://gitlab.windenergy.dtu.dk/TOPFARM/TopFarm2`, commit c50abe91d08fe8ec81fc107e88c304f9a361d348, 2022.

Gunner Chr Larsen and Pierre Elouan Réthoré. TOPFARM-a tool for wind farm optimization. In *Energy Procedia*, volume 35, pages 317–324. Elsevier Ltd, 2013. doi: 10.1016/j.egypro.2013.07.184.

Pauli Virtanen, Ralf Gommers, Travis E. Oliphant, Matt Haberland, Tyler Reddy, David Cournapeau, Evgeni Burovski, Pearu Peterson, Warren Weckesser, Jonathan Bright, Stéfan J. van der Walt, Matthew Brett, Joshua Wilson, K. Jarrod Millman, Nikolay Mayorov, Andrew R. J. Nelson, Eric Jones, Robert Kern, Eric Larson, C J Carey, İlhan Polat, Yu Feng, Eric W. Moore, Jake VanderPlas, Denis Laxalde, Josef Perktold, Robert Cimrman, Ian Henriksen, E. A. Quintero, Charles R. Harris, Anne M. Archibald, Antônio H. Ribeiro, Fabian Pedregosa, Paul van Mulbregt, and SciPy 1.0 Contributors. SciPy 1.0: Fundamental Algorithms for Scientific Computing in Python. *Nature Methods*, 17:261–272, 2020. doi: 10.1038/s41592-019-0686-2.

R. Weyler, J. Oliver, T. Sain, and J.C. Cante. On the contact domain method: A comparison of penalty and lagrange multiplier implementations. *Computer Methods in Applied Mechanics and Engineering*, 205-208:68–82, 2012. ISSN 0045-7825. doi: https://doi.org/10.1016/j.cma.2011.01.011. URL `https://www.sciencedirect.com/science/article/pii/S0045782511000120`. Special Issue on Advances in Computational Methods in Contact Mechanics.